# Protocol for a scoping review to identify and map in-service education and training materials for midwifery care in sub-Saharan Africa from 2000 to 2020

Joanne Welsh,[1] Mechthild M Gross,[1] Claudia Hanson,[2] Hashim Hounkpatin,[3] Ann-Beth Moller [4]

¹Midwifery Research and Education Unit, Hannover Medical School, Hannover, Germany
²Department of Global Public Health, Karolinska Institute, Stockholm, Sweden
³Centre de Recherche en Reproduction Humaine et en Démographie (CERRHUD), Cotonou, Benin
⁴School of Public Health and Community, Institute of Medicine, Sahlgrenska Academy, University of Gothenburg, Gothenburg, Sweden

**Correspondence to**
Ms Ann-Beth Moller;
ann-beth.moller.2@gu.se

## ABSTRACT

**Introduction** Maternal and neonatal mortality are disproportionally high in low-and middle-income countries. In 2017 the global maternal mortality ratio was estimated to be 211 per 100 000 live births. An estimated 66% of these deaths occurred in sub-Saharan Africa. Training programmes that aim to prepare providers of midwifery care vary considerably across sub-Saharan Africa in terms of length, content and quality. To overcome the shortfalls of pre-service training and support the provision of quality care, in-service training packages for providers of midwifery care have been developed and implemented in many countries in sub-Saharan Africa. We aim to identify what in-service education and training materials have been used for providers of midwifery care between 2000 and 2020 and map their content to the International Confederation of Midwives' Essential Competencies for Midwifery Practice (ICM Competencies), and the Lancet Midwifery Series Quality Maternal and Newborn Care (QMNC) framework.

**Methods and analysis** A search will be conducted for the years 2000–2020 in Cumulative Index of Nursing and Allied Health Literature, PubMed/MEDLINE, Social Sciences Citation Index, African Index Medicus and Google Scholar. A manual search of reference lists from identified studies and a hand search of literature from international partner organisations will be performed. Information retrieved will include study context, providers trained, focus of training and design of training. Original content of identified education and training materials will be obtained and mapped to the ICM Competencies and the Lancet Series QMNC.

**Ethics and dissemination** A scoping review is a secondary analysis of published literature and does not require ethical approval. This scoping review will give an overview of the education and training materials used for in-service training for providers of midwifery care in sub-Saharan Africa. Mapping the content of these education and training materials to the ICM Competencies and The Lancet Series QMNC will allow us to assess their appropriateness. Findings from the review will be reflected to stakeholders involved in the design and implementation of such materials. Additionally, findings will be published in a peer-reviewed journal, and used to inform the design and content of an in-service training package for providers of midwifery care as part of the Action Leveraging Evidence

## Strengths and limitations of this study

► This scoping review will give a comprehensive overview of the education and training materials used for in-service training for providers of midwifery care in sub-Saharan Africa.
► To our knowledge, this is the first review that aims to map the content of in-service education and training materials used for providers of midwifery care in sub-Saharan Africa to the ICM Essential Competencies and The Lancet Series Quality Maternal and Newborn Care framework.
► This scoping review may miss in-service training materials that have not been reported in journals or published in the grey literature.

to Reduce perinatal morTality and morbidity (ALERT) study, (https://alert.ki.se/) a multi-country study in Benin, Malawi, Tanzania and Uganda.

**Trial registration number** PACTR202006793783148; Post-results.

## INTRODUCTION

A transformative agenda for maternal and newborn health has been laid out as part of the Sustainable Development Goals (SDGs); 'to reduce the global maternal mortality ratio (MMR) to less than 70 per 100 000 live births by 2030' (SDG 3.1) and 'end preventable deaths of newborns, with all countries aiming to reduce neonatal mortality to at least as low as 12 per 1000 live births' (SDG 3.2).[1] Maternal and neonatal mortality are disproportionally high in low-and middle-income countries (LMICs). In 2017 the global MMR was estimated to be 211 per 100 000 live births. An estimated 66% of these deaths occurred in sub-Saharan Africa, whose regional MMR in 2017 was 542.[2] Similarly, despite a global neonatal mortality rate of 17 per 1000 live births in 2019 this rate in sub-Saharan Africa was 27 per 1000.[3] With access

**Box 1    Definition of provider of midwifery care[19 25]**

Definition of provider of midwifery care
Providers of midwifery care are competent maternal and newborn health professionals educated, trained and regulated to national and/or international standards. They provide skilled, evidence-based and compassionate care to women, newborns and families.
Providers of midwifery care:
► Promote and facilitate normal physiological, social and cultural processes throughout the childbearing continuum with a continuity of care philosophy.
► Seek to prevent and manage maternal and newborn complications.
► Consult and refer to other health services when required.
► Respect women's individual circumstances and views, providing sensitive and dignified care.

to evidence-based quality care, the majority of these deaths are preventable. Within an enabling environment, midwifery care is effective in improving care and supports the international agenda to end preventable maternal and newborn deaths.[4 5] While today many more women give birth with skilled health personnel,[6] mortality reductions have been less than anticipated.[7] This paradox might be explained in relation to the complexity of provision of high-quality maternity care which is reliant on several factors including an appropriately trained and qualified health workforce who are equitably distributed, an enabling environment which includes access to essential equipment, medications and supplies as well as social and cultural aspects of care.

The workforce providing midwifery care is diverse. A study by Hobbs et al[8] identified 102 unique cadre names from 36 LMICs who were involved in the provision of pregnancy, childbirth and postpartum care. For this review the term 'provider of midwifery care' is used. See box 1 for the definition of provider of midwifery care used in this review.

Substantial heterogeneity exists in the length, content and quality of training that different providers of midwifery care receive to prepare them for the workforce, as well as in their ability to perform obstetric and neonatal functions.[8 9] Inadequate training has been found to diminish the competence and confidence of providers of midwifery care.[10] In-service education therefore plays an important role in filling gaps that arise from inadequate pre-service education. It also acts to support continuous professional development for providers of midwifery care. A number of in-service training packages for providers of midwifery care have been developed and implemented in sub-Saharan Africa as a means to support the provision of quality maternity care.[11–14] Many in-service training packages focus on emergency obstetric care,[11–15] with few focusing on normal childbirth,[16] and dignified and respectful care.[17]

It is imperative that in-service education is evidence based, of high quality and promotes both autonomy and multidisciplinary team working. The International Confederation of Midwives[18] 'Essential Competencies

for Midwifery Practice' outline the minimum set of knowledge, skills and professional behaviour expected of an individual completing their midwifery training and joining the workforce. These essential competencies are helpful in the design and evaluation of ongoing in-service training for providers of midwifery care. Additionally, the Lancet Series Quality Maternal and Newborn Care (QMNC) framework describes the components of care that are critical to the provision of best quality maternity care and places women's views and experiences at its core.[19] The framework identifies effective practices, the organisation of care, the philosophy and values of care providers and the characteristics of care providers as essential components to the provision of quality care. The framework is designed to be relevant to any setting and as such has been recommended as a structure around which improvements in midwifery can be made globally.[20]

This review is a component of the Action Leveraging Evidence to Reduce perinatal morTality and morbidity in sub-Saharan Africa (ALERT) project (Trial registration number: PACTR202006793783148).[21] The objective of the ALERT project is to develop and evaluate a multifaceted intervention to (1) strengthen the implementation of evidence-based interventions and responsive care and, (2) reduce in-facility perinatal mortality and morbidity through a multidisciplinary approach in Benin, Malawi, Tanzania and Uganda. To achieve this, the ALERT project aims to develop and implement a co-designed in-service midwifery training package with a focus on basic intrapartum care. To inform this process this review aims to identify and map in-service education and training materials for providers of midwifery care that are available and have been used in sub-Saharan Africa since 2000.

## METHODS AND ANALYSIS
### Patient and public involvement
No patient involved

### Study design
Due to the nature of the research question a scoping review design was chosen to allow for identification and mapping of existing in-service education and training materials. This scoping review follows the steps as outlined by Arksey and O'Malley.[22] The Preferred Reporting Items for Systematic reviews and Meta-Analyses extension for Scoping Reviews (PRISMA-ScR) checklist (see online supplemental file 1) will be used to present the findings of the scoping review, and the PRISMA flow diagram (see online supplemental file 2) will be used to map out the phases of literature identification; the number of records identified, and those included and excluded with reasons given for exclusions.[23]

### Research questions
The objectives of our study are to identify, and map in-service education and training materials designed for providers of midwifery care in sub-Saharan Africa from

**Table 1** Inclusion criteria for identifying eligible studies

| | Inclusion criteria |
|---|---|
| **Population** | Any in-service training on midwifery care aimed at any health professional who provides midwifery care. |
| **Concept** | Mapping the content of in-service training materials used for providers of midwifery care to the International Confederation of Midwives' Essential Competencies and The Lancet Series Quality Maternal and Newborn Care framework. |
| **Study Design** | Quantitative studies and reports/documents that report on the implementation of in-service training materials and in-service training resources identified through searches of grey literature. |
| **Context** | In-service education in sub-Saharan Africa. |

2000 to 2020. The scoping review aims to address the following questions:

1. What in-service education and training materials are available and have been used for providers of midwifery care in sub-Saharan Africa from 2000 to 2020?
2. How does the content of these in-service education and training materials align with the International Confederation of Midwives Essential Competencies for Midwifery Practice as well as in relation to the Lancet Midwifery Series QMNC framework?

### Inclusion criteria

The research questions will be assessed, and studies will be selected specific to the following Population, Concept, Study Design and Context criteria presented in table 1. All quantitative studies and grey literature that include information on available in-service education and training materials used for providers of maternity care in sub-Saharan Africa will be included in the review. Evidence in the field of clinical care changes frequently and new guidelines and recommendations are updated when new knowledge becomes available. Some guidelines are even 'living' guidelines as evidence is constantly emerging (e.g., clinical care in relation to COVID-19). As the focus of the scoping review is on training materials based on evidence, only those developed and used after 2000 onwards will be included.

### Exclusion criteria

Articles will not be eligible for inclusion in the scoping review if:

1. The country in which the in-service education and training has taken place is not located within sub-Saharan Africa.
2. The health personnel who have undertaken the in-service education and training is not considered to be a provider of midwifery care.
3. The content of the in-service education and training does not relate to the provision of midwifery care.
4. The in-service education and training materials were developed before 2000
5. There are no details provided on the content of the education and training sessions.

### Search strategy

The search strategy will be conducted for all relevant existing literature, without language restrictions, based on search terms relating to the research questions and restricted to the years 2000–2020 using the following online bibliographic databases: Cumulative Index of Nursing and Allied Health Literature (CINAHL), PubMed/MEDLINE, Social Sciences Citation Index, African Index Medicus and Google Scholar. Grey literature searches will be performed and include organisations known to be active in global health improvement (i.e., United Nations Population Fund, World Health Organization, Johns Hopkins Program for International Education in Gynecology and Obstetrics, International Confederation of Midwives (ICM), International Federation of Gynecology and Obstetrics, International Pediatric Association). The reference list of all eligible studies will be hand-searched to identify any additional relevant studies. Examples of the preliminary search terms and strategy to be conducted in CINAHL are outlined in online supplemental file 3.

### Study selection

Following the aforementioned comprehensive search strategy, article titles and abstracts will be screened and eligibility for inclusion assessed independently by two reviewers (A-BM and JW). Any disagreements will be resolved by a third reviewer. Screened abstracts identified for inclusion will have their full texts independently screened by A-BM and JW. As the scoping review aims to map the content of in-service training materials we will not assess the quality of the studies reporting on the in-service training. All studies meeting inclusion criteria will therefore be included in the review. Full-text articles that are excluded at the screening stage will have reasons for exclusion documented. The Covidence[24] software will be used for both title/abstract and full-text screening.

### Data extraction and analysis

A data collection tool will be used to extract required information (see online supplemental file 4). Information retrieved will include study context (country and healthcare setting). Details about which type of professionals provding midwifery care were trained will be

recorded. Further data will be collected regarding the focus of the training and design of the training (formal education sessions, training facilitators, on-site training, off-site training, use of simulation, ongoing mentorship, online training). The data extraction form will contain additional fields to allow for flexibility should any other categories of training design arise that the authors had not initially considered.

Education and training materials identified in the scoping review will be sought by the reviewers. If the reviewers are unable to access the materials, they will contact the authors and ask them to share materials. This will allow original content to be mapped to the ICM Essential Competencies for Midwifery Practice[18] and the Lancet Series QMNC.[19] The tools used to map content of education and training materials can be found in online supplemental file 5. Descriptive statistics will be used to summarise the findings on how the identified in-service training materials contain content pertaining to the ICM Essential Competencies and the Lancet Series QMNC framework.

## ETHICS AND DISSEMINATION

Scoping reviews collect and examine data from existing literature and therefore do not require prior ethical approval. This scoping review will give a comprehensive overview of the education and training materials used for in-service training for providers of midwifery care in sub-Saharan Africa. To our knowledge, this is the first review that aims to map the content of education and training materials used for providers of midwifery care in sub-Saharan Africa to the ICM Essential Competencies and The Lancet Series QMNC framework. To ensure future in-service education and training materials are of quality and designed to adhere to the ICM Essential Competencies and the Lancet Series QMNC framework, findings from the scoping review will be reflected to stakeholders involved in the design and implementation of such materials and will be used to inform the development of the ALERT in-service midwifery training package. The authors aim to disseminate findings from the scoping review by publication in a peer-reviewed journal.

**Contributors** JW wrote the original draft. A-BM contributed to the conception of the study and substantively revised the manuscript. JW and A-BM developed the data extraction form. All authors JW, A-BM, MMG, CH and HH have contributed to the manuscript and read and approved the final version.

**Funding** This study is part of the ALERT project which is funded by the European Commission's Horizon 2020 (No 847824) under a call for implementation research for maternal and child health.

**Disclaimer** The contents of this article are solely the responsibility of the authors and do not reflect the views of the European Union.

**Competing interests** None declared.

**Patient consent for publication** Not required.

**Provenance and peer review** Not commissioned; externally peer reviewed.

**ORCID iD**
Ann-Beth Moller http://orcid.org/0000-0003-3581-0938

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
