## [Reviewer comments · BMJ Open]

ARTICLE DETAILS

TITLE (PROVISIONAL)	Protocol for a scoping review to identify and map in-service education and training materials for midwifery care in sub-Saharan Africa from 2000-2020
AUTHORS	Welsh, Joanne; Gross, MM; Hanson, Claudia; Hounkpatin, Hashim; Moller, Ann-Beth

VERSION 1 – REVIEW

REVIEWER	Lorraine Carroll University College Dublin Ireland
REVIEW RETURNED	17-Dec-2020

GENERAL COMMENTS	Thank you for the opportunity to review this scoping review protocol. An interesting and important topic of interest and contribution to enhance midwifery education in sub-Saharan Africa. Some comments and queries below that could further enhance the paper. 1. Page 7 Line 144 Table 1: Inclusion criteria for identifying eligible studies 'Studies and reports/documents that report on the implementation of in-service training materials and in-service training resources identified through searches of grey literature.' It is not clear if all study designs will be included e.g. quantitative and qualitative. Also see comment no. 4 below.2. Page 8 Line 159: Search strategy: Were other databases such as PubMed/Medline, EMBASE and/or Education Resources Information Centre (ERIC) and Social Science Citation Index considered? As it is a scoping review, a broader search could enhance comprehensive of retrieving relevant studies.3. Page 8 Line 170: Study Selectiona) Who will the two reviewers be? Suggest to include author initials.b) How will non-English language full text articles be screened? Will these be included?c) If there are studies with unavailable or unclear data, will authors of these studies be contacted to determine the studies eligibility to the study?d) Will two reviewers also independently screen the full text articles also?4. Page 9 Line 188: Data extraction and Analysis.
---

	Arising from comment no.1 above. If qualitative studies will be included, it is not clear how qualitative data will be synthesised. 5. Page 10 Line 200: Dissemination and Ethics Will the results from this scoping review be disseminated through peer-reviewed publication?
--	--

REVIEWER	Ntombifikile Mtshali University of KwaZulu-Natal
REVIEW RETURNED	13-Jan-2021

GENERAL COMMENTS	This is a well written scoping review protocol. The methodology and methods are well articulated and are in line with guidelines on conducting scoping reviews. Please note that under the subheading 'Data extraction and analysis, lines 186 and 187, there is the sentence that does not clearly fit under this section. It is about Quality Appraisal of selected literature. The sentence referred to is " As the scoping review aims to map the content of in-service training materials we will not assess the quality of the studies reporting on the in-service training". This statement is not about data extraction or analysis. It may read better under the 'study selection' subheading to conclude that section. This statement is in line with Arksey and O'Malley point that quality appraisal of included literature is not a requirement when performing scoping reviews. This scoping review is very important in a region that is characterised by high maternal mortality rates and quality of midwifery services that requires strengthening. It will not only benefit sub-Saharan Africa but the Region as a whole and beyond. Looking forward to the final product.
--

VERSION 1 – AUTHOR RESPONSE

Reviewer 1. comments	Authors response
Page 7 Line 144 Table 1: Inclusion criteria for identifying eligible studies 'Studies and reports/documents that report on the implementation of in-service training materials and in-service training resources identified through searches of grey literature.' It is not clear if all study designs will be included e.g. quantitative and qualitative. Also see comment no. 4 below.	Thank you for this comment. We agree that this needs clarification. We intend to include quantitative studies only. We have clarified this in the text. Quantitative studies and reports/documents that report on the implementation of in-service training materials and in-service training resources identified through searches of grey literature
Page 8 Line 159: Search strategy: Were other databases such as PubMed/Medline, EMBASE and/or Education Resources Information Centre (ERIC) and Social Science Citation Index considered? As it is a scoping review, a broader search could	Thank you for this feedback. We have considered the databases you have suggested. We have decided not to search EMBASE and Education Resources Information Centre (ERIC) as they primarily have a high income country focus. We will however expand our search to include

enhance comprehensive of retrieving relevant studies.	PubMed/Medline and Social Science Citation Index. We have reflected this in the text as follows: The search strategy will be conducted for all relevant existing literature, without language restrictions, based on search terms relating to the research questions and restricted to the years 2000-2020 using the following online bibliographic databases: Cumulative Index of Nursing and Allied Health Literature (CINAHL), PubMed/MEDLINE, Social Science Citation Index, African Index Medicus, and Google Scholar.
Page 8 Line 170: Study Selection a) Who will the two reviewers be? Suggest to include author initials. b) How will non-English language full text articles be screened? Will these be included? c) If there are studies with unavailable or unclear data, will authors of these studies be contacted to determine the studies eligibility to the study? d) Will two reviewers also independently screen the full text articles also?	Thank you for these important comments. We have addressed them as follows: a) We have added the initials of the two reviewers b) The authors are able to screen articles in English, French, Italian, Portuguese, and Spanish. Should we encounter articles written in other languages, we will seek additional translation support, which would be acknowledged in the acknowledgments section of the final paper. c) As the main aim of this scoping review is to identify in-service training materials that have been used, and then map them to the ICM competencies and QMNC Framework, we are not concerned about unavailable or unclear data, rather the information about the in-service training materials that have been used. Our exclusion criteria indicate that if no details are provided in the paper about the in-service training materials, the study will not be included. However, if the in-service training materials used are mentioned in the paper, but we are unable to access them, we will contact the authors to ask them to share the materials. We have clarified this further in the text. Education and training materials identified in the scoping review will be sought by the reviewers. If the reviewers are unable to access the materials, they will contact the authors and ask them to share materials.

	d) Yes. Full text articles will be screened independently. We have clarified this in the text: Screened abstracts identified for inclusion will have their full texts independently screened by ABM and JW.
Page 9 Line 188: Data extraction and Analysis. Arising from comment no.1 above. If qualitative studies will be included, it is not clear how qualitative data will be synthesised.	Thank you for seeking clarification over this. Qualitative studies will not be included in the synthesis.
Page 10 Line 200: Dissemination and Ethics Will the results from this scoping review be disseminated through peer-reviewed publication?	Thank you for this comment. This is an excellent point, and we do plan to share results through peer reviewed publication. We have added this information to the text. The authors aim to disseminate findings from the scoping review by publication in a peer reviewed journal.
Reviewer 2 Comments	
Please note that under the subheading 'Data extraction and analysis, lines 186 and 187, there is the sentence that does not clearly fit under this section. It is about Quality Appraisal of selected literature. The sentence referred to is " As the scoping review aims to map the content of in-service training materials we will not assess the quality of the studies reporting on the in-service training". This statement is not about data extraction or analysis. It may read better under the 'study selection' subheading to conclude that section. This statement is in line with Arksey and O'Malley point that quality appraisal of included literature is not a requirement when performing scoping reviews.	Thank you for this comment. We agree with you and have moved the statement as suggested. It can now be found under the heading "Study selection"

VERSION 2 – REVIEW

REVIEWER	Lorraine Carroll University College Dublin, Ireland
REVIEW RETURNED	22-Feb-2021

GENERAL COMMENTS	Thank you for clarity on my previous comments, and associated edits. I am pleased to recommend your paper for publication.
--